# With a Little Help from My Friend: Server-Aided Federated Learning with Partial Client Participation

**Haibo Yang**
Dept. of ECE
The Ohio State University
Columbus, OH 43210
yang.5952@osu.edu

**Peiwen Qiu**
Dept. of ECE
The Ohio State University
Columbus, OH 43210
liu.9384@osu.edu

**Prashant Khanduri**
Dept. of CS
Wayne State University
Detroit, MI 48202
khanduri.prashant@wayne.edu

**Jia Liu**
Dept. of ECE
The Ohio State University
Columbus, OH 43210
liu@ece.osu.edu

## Abstract

Although federated learning (FL) has been a prevailing distributed learning framework in recent years due to its benefits in scalability/privacy and rich applications in practice, there remain many challenges in FL system design, such as data and system heterogeneity. Notably, most existing works in the current literature only focus on addressing data heterogeneity issues (e.g., non-i.i.d. datasets across clients), while often assuming either full client or uniformly distributed client participation. However, such idealistic assumptions on client participation rarely hold in practical FL systems. It has been frequently found in FL systems that some clients may never participate in the training (aka partial/incomplete participation) due to various reasons. This motivates us to fully investigate the impacts of incomplete FL participation and develop effective mechanisms to mitigate such impacts. Toward this end, by establishing a fundamental generalization error lower bound, we first show that conventional FL is *not* PAC-learnable under incomplete participation. To overcome this challenge, we propose a new server-aided federated learning (SA-FL) framework with an auxiliary dataset deployed at the server, which is able to revive the PAC-learnability of FL under incomplete client participation. Upon resolving the PAC-learnability challenge, we further propose the SAFARI (server-aided federated averaging) algorithm that enjoys convergence guarantee and the same level of communication efficiency and privacy as state-of-the-art FL.

## 1 Introduction

Since the seminal FedAvg (federated averaging) algorithm [1], most works on FL to date are focused on addressing data heterogeneity issues [2–14]. While these algorithms have achieved varying degrees of theoretical and/or empirical success, it does not directly address the underlying challenges of client participation resulted from system-level heterogeneity. Specifically, these methods often make "ideal" assumptions on client participation in that, in each communication round, clients either all participate or are sampled *uniformly* at random in the FL training process (referred to as "full" and "uniform" client participation in this paper, respectively). However, such assumptions on client participation rarely hold for FL systems in practice due to the aforementioned system-level heterogeneity. Indeed,

Workshop on Federated Learning: Recent Advances and New Challenges, in Conjunction with NeurIPS 2022 (FL-NeurIPS'22). This workshop does not have official proceedings and this paper is non-archival.

studies have found that system-level heterogeneity could significantly affect client participation and largely degrade the learning performance of FL [15, 16]. In FL, the participation of a client is affected not only through the request/sampling from the server, but also the client's status (e.g., busy or idle, whether or not in a stable network, privacy concerns, or battery level). Thus, the client participation is *not* solely determined by the server. Even if the server enlists clients uniformly at random, the actual client participation may signficantly deviate from uniform participation. For example, it is shown in [16] that more than 30% clients never participated in FL, while only 30% of clients contribute to 81% of the total computation even if server uniformly samples clients. Exacerbating the problem is the fact that the client's status could be unstable and time-varying. Unfortunately, in the FL literature, there remains a lack of a theoretical understanding on the impacts of incomplete client participation, let alone how to mitigate such impacts on FL. This motivates us to fill this gap by fully investigating how to address the challenges of incomplete client participation in this paper. The major contributions and main results of this paper are summarized as follows:

- We first study standard FL under incomplete client participation through the lens of statistical learning theory. We prove that FL with incomplete client participation is *not* PAC-learnable by establishing a generalization error lower bound. Our analysis reveals that no learning algorithm can approach zero generalization error under incomplete participation in FL even in the limit of infinitely many training data and iterations.

- To overcome the non-PAC-learnable challenge, we propose a server-aided federated learning framework (SA-FL) to revive PAC learnability of FL under incomplete client participation. The key idea of SA-FL is to maintain an auxiliary dataset at the server that mimics the data distributions of the non participating clients. This allows the system to mitigate the deviation caused by distribution mismatch resulting as a consequence of incomplete client participation. Under mild conditions, we establish the PAC learnability of SA-FL by proving a new generalization error bound with the joint sever-client dataset.

- After showing the PAC learnability of SA-FL, we further propose an efficient training algorithm for SA-FL called SAFARI (server-aided federated averaging). By carefully designing the update coordination between the server and the clients, we show that SAFARI achieves an $\mathcal{O}(1/\sqrt{KT})$ convergence rate.

## 2 PAC learnability analysis for FL with incomplete client participation

The goal of FL is to minimize the following loss function $F(\mathbf{x}) = \mathbb{E}_{i \sim \mathcal{P}}[F_i(\mathbf{x})]$, where $F_i(\mathbf{x}) \triangleq \mathbb{E}_{\xi \sim P_i}[f_i(\mathbf{x}, \xi)]$. Here, $\mathcal{P}$ represents the distribution of the entire client population, $\mathbf{x} \in \mathbb{R}^d$ is the model parameter, $F_i(\mathbf{x})$ represents the local loss function at client $i$, and $P_i$ is the underlying distribution of the local dataset at client $i$. In general, $P_i \neq P_j$, if $i \neq j$ due to data heterogeneity. However, the loss function $F(\mathbf{x})$ or full gradient $\nabla F(\mathbf{x})$ can not be directly computed as the exact distribution of data is unknown in general. Instead, one often considers the following empirical risk minimization (ERM) problem in the form of finite-sum:

$$\min_{\mathbf{x} \in \mathbb{R}^d} \hat{F}(\mathbf{x}) = \sum_{i \in [M]} \alpha_i \hat{F}_i(\mathbf{x}), \tag{1}$$

where $\hat{F}_i(\mathbf{x}) \triangleq \frac{1}{|S_i|} \sum_{\xi \in S_i} f_i(\mathbf{x}, \xi)$. Here, $M$ is the total number of clients, $S_i$ is the local dataset with cardinality $|S_i|$, which is i.i.d. and sampled from distribution $P_i$, $\alpha_i = \frac{|S_i|}{\sum_{j \in [M]} |S_j|}$ (hence $\sum_{i \in [M]} \alpha_i = 1$). For ease of presentation, we consider a balanced dataset case: $\alpha_i = \frac{1}{M}, \forall i \in [M]$. Next, we state several basic definitions from statistical learning theory [17] that are needed here.

**Definition 1** (Generalization Error and Empirical Error). *Given a hypothesis $h \in \mathcal{H}$, a target concept $f$, an underlying distribution $\mathcal{D}$ and a dataset $S$ i.i.d. sampled from $\mathcal{D}$ ($S \sim \mathcal{D}$), the generalization error and empirical error of $h$ are defined as follows $\mathcal{R}_{\mathcal{D}}(h) = \mathbb{E}_{(x,y) \sim \mathcal{D}} l(h(x), f(x))$ and $\hat{\mathcal{R}}_D(h) = \frac{1}{|S|} \sum_{i \in S} l(h(x_i), f(x_i))$, where $l(\cdot)$ is some valid loss function.*

**Definition 2** (Optimal Hypothesis). *For a distribution $\mathcal{D}$ and a dataset $S \sim \mathcal{D}$, we define $h_{\mathcal{D}}^* = \underset{h \in \mathcal{H}}{\mathrm{argmin}} \mathcal{R}_{\mathcal{D}}(h)$ and $\hat{h}_{\mathcal{D}}^* = \underset{h \in \mathcal{H}}{\mathrm{argmin}} \hat{\mathcal{R}}_{\mathcal{D}}(h)$.*

**Definition 3** (Excess Error). *For hypothesis $h$ and distribution $\mathcal{D}$, the excess error and excess empirical error are defined as $\varepsilon_{\mathcal{D}}(h) = \mathcal{R}_{\mathcal{D}}(h) - \mathcal{R}_{\mathcal{D}}(h_{\mathcal{D}}^*)$, and $\hat{\varepsilon}_{\mathcal{D}}(h) = \hat{\mathcal{R}}_{\mathcal{D}}(h) - \hat{\mathcal{R}}_{\mathcal{D}}(\hat{h}_{\mathcal{D}}^*)$, respectively.*

## 2.1 FL under incomplete client participation

With the above notations, we consider the FL under incomplete client participation. Consider an FL system with $M$ clients in total. We let $P$ denote the underlying joint distribution of the entire system, which can be decomposed into the summation of the local distributions at each client, i.e., $P = \sum_{i \in [M]} \lambda_i P_i$, where $\lambda_i > 0$ and $\sum_{i \in [M]} \lambda_i = 1$. We assume that each client $i$ has $n$ training samples i.i.d. drawn from $P_i$, i.e., $|S_i| = n, \forall i \in [M]$. Then, $S = \{(x_i, y_i), i \in [M \times n]\}$ can be viewed as the dataset i.i.d. sampled from the joint distribution $P$. We consider an incomplete client participation setting, where $m \in [0, M)$ clients participate in the FL training as a result of some client sampling process $\mathcal{F}$. We let $\mathcal{F}(S)$ represent the data ensemble actually used in the training and $\mathcal{D}$ denote the underlying distribution corresponding to $\mathcal{F}(S)$. For convenience, we define the notion $\alpha = \frac{m}{M}$ as the *FL system capacity* (i.e., only $m$ clients participate in the training).

For FL with incomplete client participation, we establish the following fundamental performance limit of any learner in general. For simplicity, we use binary classification with zero-one loss here. We state the following impossibility result in Theorem 1 in terms of PAC learnability:

**Theorem 1** (Impossibility Theorem)**.** *Let $\mathcal{H}$ be a non-trivial hypothesis space and $\mathcal{L}$ : $(\mathcal{X}, \mathcal{Y})^{(m \times n)} \to \mathcal{H}$ be the learner for an FL system. There exists a client participation process $\mathcal{F}$, a distribution $P$, and a target function $f \in \mathcal{H}$ with $\min_{h \in \mathcal{H}} \mathcal{R}_P(h, f) = 0$, such that*

$$\mathbb{P}_{S \sim P}\left[\mathcal{R}_P(\mathcal{L}(\mathcal{F}(S)), f) > \frac{1 - \alpha}{8}\right] > \frac{1}{20}. \tag{2}$$

*Proof Sketch.* The proof is based on the method of induced distributions in [17–19]. We first show that the learnability of an FL system is equivalent to that of a system that arbitrarily selects $mn$ out of $Mn$ samples in centralized learning. Then, for any learning algorithm, there exists a distribution $P$ such that dataset $\mathcal{F}(S)$ resulting from incomplete participation and seen by the algorithm is always distributed identically for any target functions. Thus, no algorithm can learn a better predictor than random guessing. Due to space limitation, we relegate the full proof to supplementary material. $\square$

Given the system capacity $\alpha \in (0, 1)$, the above theorem characterizes the worst-case scenario for FL with incomplete client participation. It shows that for any learner (i.e., algorithm) $\mathcal{L}$, there exists a bad client participation process $\mathcal{F}$ and distributions $P_i, i \in [M]$ over target function $f$, for which the error of the hypotheses returned by $\mathcal{L}$ is constant with non-zero probability. In other words, FL with incomplete client participation is *not PAC-learnable*. One interesting observation here is that the lower bound is independent of the number of samples per client $n$. This indicates that even if each client has infinitely many samples ($n \to \infty$), it is impossible to have a zero-generation-error learner under the incomplete client participation situation ($\alpha \in (0, 1)$). Note that this fundamental result relies on two conditions: *heterogeneous* dataset and *arbitrary* client participation. Under these two conditions, there exists a worst-case scenario where the underlying distribution $\mathcal{D}$ of the participating data $S_\mathcal{D} = \mathcal{F}(S)$ deviates from the ground-truth $P$, thus yielding a non-vanishing error. This result also sheds light on how to motivate client participation in FL effectively and efficiently: the participating client's data should be comprehensive enough to model the complexity of the joint distribution $P$ to close the gap between $\mathcal{D}$ and $P$.

Note that this result is not contradictory to previous works where the convergence of FedAvg is guaranteed since this theorem is not applicable for homogeneous (i.i.d.) datasets or uniformly random client participation. As mentioned earlier, most of the existing works rely on at least one of these two assumptions. However, none of these two assumptions hold for FL with incomplete client participation. Theorem 1 naturally leads to an important open question: *How to make FL with incomplete client participation PAC-learnable?* Toward this end, we propose a new server-aided federated learning framework (SA-FL) in the next subsection.

## 2.2 Server-aided federated learning (SA-FL)

We consider the same FL problem with incomplete client participation, which induces a dataset $S_\mathcal{D} \sim \mathcal{D}$, with cardinality $n_S$ and $\mathcal{D} \neq P$. In SA-FL, the server is equipped with an auxiliary dataset $T$ of cardinality $n_T$, which is i.i.d. sampled from distribution $P$. As a result, the learning process is to minimize $\mathcal{R}_P(h)$ by utilizing $(\mathcal{X}, \mathcal{Y})^{n_T + n_S}$ to learn a hypothesis $h \in \mathcal{H}$. For notional clarity, we assume the joint dataset $S_Q = (S_\mathcal{D} \cup T) \sim Q$ with cardinality $n_T + n_S$ for some distribution $Q$.

The intuition of SA-FL is to utilize dataset $T$ as a vehicle to correct potential distribution derivations due to incomplete client participation. By doing so, the server steers the learning by a small number of representative data, while the clients aid the learning by federation to leverage the huge amount of privately decentralized data ($n_S \gg n_T$). For SA-FL to be practical, it is highly desirable that the server only needs to maintain a small dataset from distribution $P$. Note that the assumption of access to this dataset is not restrictive since such datasets are already available in many FL systems: although not always necessary for training, an auxiliary dataset is often needed for defining FL tasks (e.g., simulation prototyping) before training and model checking after training (e.g., quality evaluation and sanity checking) [20, 21]. Also, obtaining an auxiliary dataset is affordable since the number of data points required is relatively small (of the order of hundreds, see our experimental results), and hence the cost is low. Then, SA-FL can be easily achieved or even with manually labelled data thanks to its small size. It is also worth noting that many works use such auxiliary datasets in FL for security [22], incentive design [23], and knowledge distillation [24]. Before deriving the generalization error bound for SA-FL, we state the following assumption and definition.

**Assumption 1** (Noise Condition). *Suppose that $h_P^*$ and $h_Q^*$ exist. There exist $\beta_P, \beta_Q \in [0, 1]$ and $\alpha_P, \alpha_Q > 0$ such that*

$$\mathbb{P}_{x \sim P}(h(x) \neq h_P^*(x)) \leq \alpha_P [\varepsilon_P(h)]^{\beta_P}, \tag{3}$$

$$\mathbb{P}_{x \sim Q}(h(x) \neq h_Q^*(x)) \leq \alpha_q [\varepsilon_Q(h)]^{\beta_Q}. \tag{4}$$

This assumption is a traditional noise model known as the Bernstein class condition, which has been widely used in the literature [25–27].

**Definition 4** (($\alpha, \beta$)-Positively-Related). *Distributions $P$ and $Q$ are said to be ($\alpha, \beta$)-positively-related if there exist constants $\alpha \geq 0$ and $\beta \geq 0$ such that*

$$|\varepsilon_P(h) - \varepsilon_Q(h)| \leq \alpha [\varepsilon_Q(h)]^{\beta}, \forall h \in \mathcal{H}. \tag{5}$$

Definition 1 specifies a stronger constraint between distributions $P$ and $Q$. It indicates that the difference of excess error for one hypothesis $h \in \mathcal{H}$ between $P$ and $Q$ is bounded by the excess error of $Q$ in some exponential form. With the above assumption and definition, we have the following generation error bound:

**Theorem 2** (Generalization Error Bound for SA-FL). *For an SA-FL system with arbitrary system and data heterogeneity, if distributions $P$ and $Q$ satisfy Assumption 1 and are ($\alpha, \beta$)-positively-related, then with probability at least $1 - \delta$ for any $\delta \in (0, 1)$, it holds that*

$$\varepsilon_P(\hat{h}_Q^*) = \widetilde{\mathcal{O}} \left( \left( \frac{d_\mathcal{H}}{n_T + n_S} \right)^{\frac{1}{2 - \beta_Q}} + \left( \frac{d_\mathcal{H}}{n_T + n_S} \right)^{\frac{\beta}{2 - \beta_Q}} \right), \tag{6}$$

*where $d_\mathcal{H}$ denotes the finite VC dimension for hypotheses class $\mathcal{H}$, and parameters $\{P, Q, n_T, n_S, \beta, \beta_Q\}$ are defined the same as before.*

It is known that (see, e.g., [27]) the generalization error bound of centralized learning is (hiding logarithmic factors) $\widetilde{\mathcal{O}}((\frac{1}{n})^{\frac{1}{2 - \beta_Q}})$ with $n$ samples in total and noise parameter $\beta_Q$. Note that when $\beta \geq 1$, the first term in Eq. (6) dominates. Hence, Theorem 2 implies that the generalization error bound for SA-FL *matches* that of centralized learning (with dataset size $n_T + n_S$). Meanwhile, compared with solely training on server's dataset $T$, SA-FL exhibits an improvement from $\widetilde{\mathcal{O}}((\frac{1}{n_T})^{\frac{1}{2 - \beta_Q}})$ to $\widetilde{\mathcal{O}}((\frac{1}{n_T + n_S})^{\frac{\beta}{2 - \beta_Q}})$. This highlights the benefit of collaboration from the clients.

Note that SA-FL shares some similarity with the domain adaptation problem, where the learning is on $Q$ but the results will be adapted to $P$. In what follows, we offer some deeper insights between the two by answering two key questions: *1) What is the difference between SA-FL and domain adaptation (a.k.a. transfer learning)?* and *2) Why is SA-FL from $Q$ to $P$ PAC-learnable, but FL from $D$ to $P$ with incomplete client participation not PAC-learnable (as indicated in Theorem 1)?*

To answer these questions, we illustrate the distribution relationships for domain adaptation and federated learning, in Fig. 1, respectively. In domain adaptation, the target $P$ and source $Q$ distributions often have overlapping support but there also exists *distinguishable difference*. In contrast, the two distributions $P$ and $Q$ in FL happen to share exactly the *same support* with different density, since $Q$

is a *mixture* of $D$ and $P$. As a result, the known bounds in domain adaptation (or transfer learning) are pessimistic in FL. For example, the $dist(P, Q)$ in $d_{\mathcal{A}}$-divergence and $\mathcal{Y}$-divergence both have non-negligible gaps when applied to FL. Here in Theorem 2, we provide a generalization error bound in terms of the total sample size $n_T + n_S$, thus showing the benefit of SA-FL.

Moreover, for FL, only the auxiliary dataset $T \overset{i.i.d.}{\sim} P$ is directly available for the server. The clients' datasets could be used in FL training, but they are not directly accessible due to privacy constraints. Therefore, previous methods in domain adaptation (e.g., importance weights-based methods in covariate shift adaptation [28, 29]) are *not* applicable since they require the knowledge of density ratio between training and test datasets.

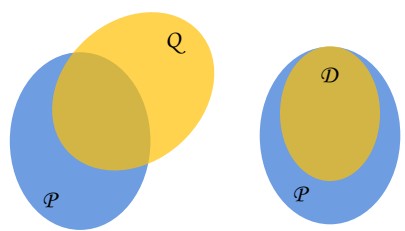

Figure 1: Diagram of distribution for domain adaptation and federated learning

The key difference between FL and SA-FL lies in relations among $D, P$ and $Q$. For FL, the distance between $D$ and $P$ under incomplete participation could be large due to system and data heterogeneity in the worst-case. More specifically, the support of $D$ could be narrow enough to miss some part of $P$, resulting in non-vanishing error as indicated in Theorem 1. For SA-FL, distribution $Q$ is a mixture of $P$ and $D$ ($Q = \lambda_1 D + \lambda_2 P$, with $\lambda_1, \lambda_2 \geq 0, \lambda_1 + \lambda_2 = 1$), thus having the same support with $P$. Hence, under Assumption 4, the PAC-learnability is guaranteed.

Although we provide a promising bound to show the PAC-learnability of SA-FL in Theorem 2, the superiority of SA-FL over training solely with dataset $T$ in server (i.e., $\widetilde{\mathcal{O}}((\frac{1}{n_T})^{\frac{1}{2-\beta_P}})$) is not always guaranteed as $\beta \to 0$ (i.e., $Q$ becomes increasingly different from $P$). In what follows, we reveal under what conditions between $P$ and $Q$ could SA-FL perform *no worse than* centralized learning in terms of generalization error.

**Theorem 3** (SA-FL Being No Worse Than Centralized Learning). *Consider an SA-FL system with arbitrary system and data heterogeneity. If Assumption 1 holds and additionally $\hat{\mathcal{R}}_P(\hat{h}_Q^*) \leq \hat{\mathcal{R}}_P(h_Q^*)$ and $\varepsilon_P(h_Q^*) = \mathcal{O}(\mathcal{A}(n_T, \delta))$, then with probability at least $1 - \delta$ for any $\delta \in (0, 1)$, it holds that $\varepsilon_P(\hat{h}_Q^*) = \widetilde{\mathcal{O}}\left((d_{\mathcal{H}}/n_T)^{\frac{1}{2-\beta_P}}\right)$, where $\mathcal{A}(n_T, \delta) = \frac{d_{\mathcal{H}}}{n_T} \log(\frac{n_T}{d_{\mathcal{H}}} + \frac{1}{n_T} \log(\frac{1}{\delta}))$, and other parameters are as defined in Theorem 2.*

Here, we remark that $\varepsilon_P(h_Q^*) = \mathcal{O}(\mathcal{A}(n_T, \delta))$ is a weaker condition than the $\varepsilon_P(h_Q^*) = 0$ condition and the covariate shift assumption ($P_{Y|X} = Q_{Y|X}$) used in the transfer learning literatures [30, 31]. Together with the condition $\hat{\mathcal{R}}_P(\hat{h}_Q^*) \leq \hat{\mathcal{R}}_P(h_Q^*)$, the following intermediate result holds: $\hat{\mathcal{R}}_P(\hat{h}_Q^*) - \hat{\mathcal{R}}_P(h_P^*) = \mathcal{O}(A(n_T, \delta))$ (see Lemma 2 in the supplementary material). Intuitively, this states that "if $P$ and $Q$ share enough similarity, then the difference of excess empirical error between $\hat{h}_Q^*$ and $h_P^*$ on $P$ can be bounded." Thus, the excess error of $\hat{h}_Q^*$ shares the same upper bound as that of $\hat{h}_P^*$ in centralized learning. Therefore, Theorem 3 implies that, under mild conditions, SA-FL guarantees the same generalization error upper bound as that of centralized learning with dataset $T$ (to see this, set $n_S = 0$ and $\beta = 0$ in Eq. (6)), hence being "no worse than" centralized learning with dataset $T$.

## 3 The SAFARI algorithm

In the previous section, we showed that SA-FL under arbitrary system heterogeneity is PAC learnable and its superiority to standard FL is guaranteed under mild conditions. This indicates the existence of an algorithm to achieve PAC learnability for SA-FL. However, it remains to design some efficient algorithms for SA-FL with a comparable level of communication overhead as conventional FL. In this section, we propose our SAFARI (server-aided federated averaging) algorithm for SA-FL and characterize its convergence guarantees.

As shown in Algorithm 1, SAFARI iteratively performs the following three steps: 1) Server samples a subset of clients as in conventional FL and synchronize the latest global model $\mathbf{x}_t$ with each participating clients (Line 3). 2) Each participating client and the server train the model based on local dataset (Lines 4-8). Specifically, client initializes its local model with $\mathbf{x}_t$ and then performs $K$

---

**Algorithm 1** The SAFARI Algorithm for SA-FL.

---

1: Initialize $\mathbf{x}_0$.
2: **for** $t = 0, \cdots, T-1$ **do**
3:     The server samples a subset $S_t$ of clients with $|S_t| = n$ and send current model $x_t$.
4:     **for** Each client $i \in S_t$ **do**
5:         Synchronization: $\mathbf{x}_{t,0}^i = \mathbf{x}_t$.
6:         Local updates: for $k = 0, ..., K-1$:     $\mathbf{x}_{t,k+1}^i = \mathbf{x}_{t,k}^i - \eta \nabla F_i(\mathbf{x}_{t,k}^i, \xi_{t,k}^i)$.
7:         Send $\Delta_t^i = - \sum_{k=0}^{K-1} \nabla F_i(\mathbf{x}_{t,k}^i, \xi_{t,k}^i)$ to server.
8:     **end for**
9:     **for** Server **do**
10:         Local updates: for $k = 0, ..., K-1$:     $\mathbf{x}_{t,k+1}^0 = \mathbf{x}_{t,k}^0 - \eta \nabla F(\mathbf{x}_{t,k}^0, \xi_{t,k}^0)$,
                $\Delta_t^0 = - \sum_{k=0}^{K-1} \nabla F(\mathbf{x}_{t,k}^0, \xi_{t,k}^0)$.
11:         Receive $\Delta_t^i, i \in S_t$, and normalize it:     $\hat{\Delta}_t^i = c_t \frac{\Delta_t^i - \Delta_t^0}{\|\Delta_t^i - \Delta_t^0\|}$.
12:         Server Update:
                $\mathbf{x}_{t+1} = \mathbf{x}_t + \eta \left( \Delta_t^0 + \frac{1}{|S_t|} \sum_{i \in S_t} \hat{\Delta}_t^i \right)$.
13:     **end for**
14: **end for**

---

local steps by the stochastic gradient descent method. Then, each client sends its locally accumulated update $\Delta_t^i$ back to the server. Note the server simultaneously takes $K$ local steps based on its auxiliary dataset (in Line 10). 3) Server aggregates and updates the global model (Lines 11-12). Upon receiving the local update $\Delta_t^i$, the server normalizes and rescales it by a hyper-parameter $c_t$. Then, the server updates the global model by aggregating the normalized update $\hat{\Delta}_t^i$ and the server's update $\Delta_t^0$ based on its own auxiliary dataset. Compared to FedAvg [1] in FL, SAFARI shares the same communication and computation process from the client's perspective. Hence, it enjoys the same level of communication efficiency and privacy benefits.

**Assumption 2.** *(L-Lipschitz Continuous Gradient) There exists a constant $L > 0$, such that $\|\nabla F(\mathbf{x}) - \nabla F(\mathbf{y})\| \leq L\|\mathbf{x} - \mathbf{y}\|, \forall \mathbf{x}, \mathbf{y} \in \mathbb{R}^d$.*

**Assumption 3.** *(Unbiased Stochastic Gradient with Bounded Variance) The stochastic gradient is unbiased, i.e., $\mathbb{E}[\nabla F(\mathbf{x}, \xi)] = \nabla F(\mathbf{x})$ and $\mathbb{E}[\|\nabla F(\mathbf{x}, \xi) - \nabla F(\mathbf{x})\|^2] \leq \sigma^2$.*

With the assumptions above, we are now in the position to analyze the convergence of SAFARI.

**Theorem 4** (Convergence Rate for SAFARI ). *Under Assumptions 2 and 3, let constant learning rate $\eta$ satisfy $(\frac{1}{2} - 4LK\eta - 20K(L + 4KL^3\eta)\eta^2) > 0$. Then, the sequence $\{\mathbf{x}_t\}$ generated by the SAFARI algorithm satisfies:*

$$\frac{1}{T} \sum_{t=0}^{T-1} \mathbb{E}\|\nabla F(\mathbf{x}_t)\|^2 \leq \frac{1}{c} \left[ \frac{F(\mathbf{x}_0) - F(\mathbf{x}^*)}{\eta K T} \right] + \frac{1}{c} \left[ \left( 5KL\eta^2 + 20K^2L^3\eta^3 + 2L\eta \right) \sigma^2 \right]$$

$$+ \frac{1}{c} \left[ \left( \frac{1}{K^2} + \frac{L\eta}{K} \right) \frac{1}{T} \sum_{t=0}^{T-1} c_t^2 \right],$$

*where $c$ is a constant and $\mathbf{x}^*$ denotes an optimal solution.*

Theorem 4 implies an $\mathcal{O}(1/T)$ convergence rate to a neighborhood of a stationary point. Furthermore, by choosing parameters $\{c_t\}$ and the learning rate $\eta$ appropriately, we have the following convergence rate to a stationary point:

**Corollary 1.** *If $\sum_{t=0}^{T-1} c_t^2$ is bounded and learning rate $\eta = \frac{1}{\sqrt{KT}}$, the convergence rate of SAFARI is:*

$$\mathcal{O} \left( \frac{1}{K^{1/2}T^{1/2}} + \frac{1}{T} + \frac{1}{K^2 T} + \frac{1}{K^{3/2}T^{3/2}} \right).$$

In Theorem 4 and Corollary 1, we show the convergence guarantee of SAFARI under no extra assumptions on the data and system heterogeneity (client participation), which corroborates the learnability

analysis in Section 2. Hence, in the worst-case scenarios, convergence rate $\mathcal{O}(1/(K^{1/2}T^{1/2}))$ is achieved for sufficiently large $T$ and $K \leq T$. In comparison, a non-vanishing error term emerges consistently for the same setting in FL [32]. This verifies the superiority of SA-FL over conventional FL with incomplete client participation. Note that Corollary 1 requires a convergent series $\{c_t^2\}$. This can be relaxed to $\sum_{t=0}^{T-1} c_t^2 = \mathcal{O}(\min\{KT, K^{3/2}T^{1/2}\})$ to maintain the same $\mathcal{O}(1/(K^{1/2}T^{1/2}))$ rate. It can be readily verified that $p$-series ($c_t = t^{-p}$) satisfies the condition.

## 4   Conclusion

Different from previous works that considered either full or uniform client participation scenarios in federated learning (FL), we considered in this paper a more practical scenario in FL with incomplete client participation. By establishing a fundamental generalization error lower bound, we first showed that conventional FL is *not* PAC-learnable under incomplete client participation. To overcome this challenge, we proposed a new server-aided federated learning (SA-FL) framework with an auxiliary dataset deployed at the server, which is able to revive the PAC-learnability of FL under incomplete client participation. Upon resolving the PAC-learnability challenge, we proposed a new SAFARI (server-aided federated averaging) algorithm that enjoys convergence guarantee and the same level of communication efficiency and privacy protection as conventional FL.

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

## A   Proofs

**Theorem 1** (Impossibility Theorem). *Let $\mathcal{H}$ be a non-trivial hypothesis space and $\mathcal{L}$ : $(\mathcal{X}, \mathcal{Y})^{(m \times n)} \to \mathcal{H}$ be the learner for an FL system. There exists a client participation process $\mathcal{F}$, a distribution $P$, and a target function $f \in \mathcal{H}$ with $\min_{h \in \mathcal{H}} \mathcal{R}_P(h, f) = 0$, such that*

$$\mathbb{P}_{S \sim P} \left[ \mathcal{R}_P(\mathcal{L}(\mathcal{F}(S)), f) > \frac{1 - \alpha}{8} \right] > \frac{1}{20}. \tag{2}$$

*Proof.* Denote $S$ the dataset with size $Mn$ i.i.d. sampled from distribution $P$, $\mathcal{F}(\cdot)$ the sampling process of FL system, and $\bar{S} = \mathcal{F}(S)$ the training dataset selected by FL system with size $mn$. Consider a distribution $P$ with support on only two points $\{x_1, x_2\}$ such that $\mathbb{P}_P(x_1) = 1 - 4\epsilon$ and $\mathbb{P}_P(x_2) = 4\epsilon$ with $\epsilon = \frac{1-\alpha}{8}$.

First we show that the rare points $x_2$ appears at most $(1 - \alpha)Mn$ times with constant probability. Let $\hat{s}$ be the number of $x_2$ points in $S$, then $\hat{s} \sim \mathbb{B}(Mn, \epsilon)$ is a binomial random variable. By the Chernoff bound,

$$\mathbb{P}[\hat{s} \geq (1 - \alpha)Mn] = \mathbb{P}[\hat{s} \geq (1 + 1)4\epsilon Mn] \leq e^{-\frac{4\epsilon Mn}{3}} = e^{-\frac{(1-\alpha)Mn}{6}} \leq e^{-\frac{1}{6}} \leq \frac{17}{20}.$$

So $\mathbb{P}[\hat{s} < (1 - \alpha)Mn] > \frac{3}{20}$.

Next, we consider the following sampling process with dataset $S = \{(x_1', f(x_1')), \ldots, (x_{M \times n}', f(x_{M \times n}'))\}$: choosing as many data $(x_i', f(x_i')), i \in [mn]$ such that $x_i' = x_1$ as possible to form the training set $\bar{S}$. Let $f_1, f_2 \in \mathcal{H}$ be two target functions whose existence is guaranteed by the non-trivial definition of $\mathcal{H}$ and $f_1(x_1) = f_2(x_1), f_1(x_2) = -f_2(x_2)$, and $\mathcal{S}$ be the set of all datasets in $(\mathcal{X}, \mathcal{Y})^{(M \times n)}$ such that $\hat{s} < (1 - \alpha)MN$.

Let $\mathcal{R}(h_s, f) = \mathbb{P}_P[\mathcal{L}(\mathcal{F}(S))(x) \neq f_1(x) \cap x \neq x_1]$, the following holds for these two target functions $f_1$ and $f_2$:

$$\begin{aligned} \mathcal{R}(h_s, f_1) + \mathcal{R}(h_s, f_2) &= \mathbb{P}_P[\mathcal{L}(\mathcal{F}(S))(x) \neq f_1(x) \cap x \neq x_1] + \mathbb{P}_P[\mathcal{L}(\mathcal{F}(S))(x) \neq f_2(x) \cap x \neq x_1] \\ &= \mathbf{1}_{\mathcal{L}(\mathcal{F}(S))(x_1) \neq f_1(x_1)} \mathbb{P}(x_2) + \mathbf{1}_{\mathcal{L}(\mathcal{F}(S))(x_1) \neq f_2(x_2)} \mathbb{P}(x_1) \\ &= 4\epsilon. \end{aligned}$$

The above result hold in expectation since it holds for any $S \in \mathcal{S}$. Hence, there exists a target function $f \in \mathcal{H}$ such that $\mathbb{E}_{S \in \mathcal{S}} \mathcal{R}(h_s, f) \geq 2\epsilon$. Note $\mathcal{R}(h_s, f) \leq \mathbb{P}(x \neq x_1) = 4\epsilon$, then by decomposing the expectation into two parts we obtain:

$$\begin{aligned} 2\epsilon \leq \mathbb{E}_{S \in \mathcal{S}} \mathcal{R}(h_s, f) &= \sum_{S : \mathcal{R}(h_s, f) \geq \epsilon} \mathcal{R}(h_s, f) \mathbb{P}[\mathcal{R}(h_s, f)] + \sum_{S : \mathcal{R}(h_s, f) < \epsilon} \mathcal{R}(\mathcal{R}(h_s, f) \mathbb{P}[\mathcal{R}(h_s, f)] \\ &\leq 4\epsilon \mathbb{P}_{S \in \mathcal{S}}[\mathcal{R}(h_s, f) \geq 4\epsilon] + \epsilon(1 - \mathbb{P}_{S \in \mathcal{S}}[\mathcal{R}(h_s, f) \geq \epsilon]) \\ &= \epsilon + 3\epsilon \mathbb{P}_{S \in \mathcal{S}}[\mathcal{R}(h_s, f) \geq \epsilon]. \end{aligned}$$

That is,

$$\mathbb{P}_{S \in \mathcal{S}}[\mathcal{R}(h_s, f) \geq \epsilon] \geq \frac{1}{3}.$$

Note $\mathcal{R}(h_s, f) = \mathbb{P}_P[\mathcal{L}(\mathcal{F}(S))(x) \neq f_1(x) \cap x \neq x_1] \leq \mathcal{R}(\mathcal{L}(\mathcal{F}(S))) = \mathbb{P}_P[\mathcal{L}(\mathcal{F}(S))(x) \neq f_1(x)]$, then we have the final results:

$$\begin{aligned} \mathbb{P}_{S \sim P}[\mathcal{R}_P(\mathcal{L}(\mathcal{F}(S)), f) \geq \epsilon] &\geq \mathbb{P}_{S \sim P}[\mathcal{R}(h_s, f) \geq \epsilon] \\ &\geq \mathbb{P}_{S \in \mathcal{S}}[\mathcal{R}(h_s, f) \geq \epsilon] \mathbb{P}[S \in \mathcal{S}] \\ &> \frac{1}{3} \frac{3}{20} = \frac{1}{20}. \end{aligned}$$

$\square$

**Theorem 2** (Generalization Error Bound for SA-FL). *For an SA-FL system with arbitrary system and data heterogeneity, if distributions $P$ and $Q$ satisfy Assumption 1 and are $(\alpha, \beta)$-positively-related, then with probability at least $1 - \delta$ for any $\delta \in (0, 1)$, it holds that*

$$\varepsilon_P(\hat{h}_Q^*) = \widetilde{\mathcal{O}} \left( \left( \frac{d_{\mathcal{H}}}{n_T + n_S} \right)^{\frac{1}{2 - \beta_Q}} + \left( \frac{d_{\mathcal{H}}}{n_T + n_S} \right)^{\frac{\beta}{2 - \beta_Q}} \right), \tag{6}$$

where $d_{\mathcal{H}}$ denotes the finite VC dimension for hypotheses class $\mathcal{H}$, and parameters $\{P, Q, n_T, n_S, \beta, \beta_Q\}$ are defined the same as before.

*Proof.*

$$\begin{aligned}
\varepsilon_P(\hat{h}_Q^*) &= \mathcal{R}_P(\hat{h}_Q^*) - \mathcal{R}_P(h_P^*) \\
&= [\mathcal{R}_P(\hat{h}_Q^*) - \mathcal{R}_P(h_P^*) - (\mathcal{R}_Q(\hat{h}_Q^*) - \mathcal{R}_Q(h_Q^*))] + \mathcal{R}_Q(\hat{h}_Q^*) - \mathcal{R}_Q(h_Q^*) \\
&\leq |\varepsilon_P(\hat{h}_Q^*) - \varepsilon_Q(\hat{h}_Q^*)| + \varepsilon_Q(\hat{h}_Q^*) \\
&\leq \alpha \varepsilon_Q(\hat{h}_Q^*)^\beta + \varepsilon_Q(\hat{h}_Q^*).
\end{aligned}$$

Combining with Lemma 1, the proof is complete.

**Lemma 1** (Auxiliary Lemma [25, 26, 30, 31]). *For any $m \in \mathbb{N}$ and $\delta \in (0,1)$, define $A(m, \delta) = \frac{d_{\mathcal{H}}}{m} \log(\frac{m}{d_{\mathcal{H}}} + \frac{1}{m} \log(\frac{1}{\delta}))$ With probability at least $1 - \delta$, $\forall h, \hat{h} \in \mathcal{H}$,*

$$\mathcal{R}(h) - \mathcal{R}(\hat{h}) \leq \hat{\mathcal{R}}(h) - \hat{\mathcal{R}}(\hat{h}) + c\sqrt{\min\{\mathbb{P}_S(h \neq \hat{h}), \hat{\mathbb{P}}_S(h \neq \hat{h})\} A(m, \delta)} + c A(m, \delta),$$

$$\frac{1}{2} \mathbb{P}_S(h \neq \hat{h}) - c A(m, \delta) \leq \hat{\mathbb{P}}_S(h \neq \hat{h}) \leq 2 \mathbb{P}_S(h \neq \hat{h}) + c A(m, \delta),$$

$$\varepsilon_Q(\hat{h}_Q^*) = [A(m, \delta)]^{\frac{1}{2 - \beta_Q}},$$

*where $\mathbb{P}_S(\cdot) = \mathbb{E}[\hat{\mathbb{P}}_S(\cdot)]$, $S$ is the i.i.d. dataset with size $m$ drawn form distribution $Q$, $c \in (0, \infty)$ is a constant.*

$\square$

**Theorem 3** (SA-FL Being No Worse Than Centralized Learning). *Consider an SA-FL system with arbitrary system and data heterogeneity. If Assumption 1 holds and additionally $\hat{\mathcal{R}}_P(\hat{h}_Q^*) \leq \hat{\mathcal{R}}_P(h_Q^*)$ and $\varepsilon_P(h_Q^*) = \mathcal{O}(\mathcal{A}(n_T, \delta))$, then with probability at least $1 - \delta$ for any $\delta \in (0, 1)$, it holds that $\varepsilon_P(\hat{h}_Q^*) = \widetilde{\mathcal{O}}\left((d_{\mathcal{H}}/n_T)^{\frac{1}{2 - \beta_P}}\right)$, where $\mathcal{A}(n_T, \delta) = \frac{d_{\mathcal{H}}}{n_T} \log(\frac{n_T}{d_{\mathcal{H}}} + \frac{1}{n_T} \log(\frac{1}{\delta}))$, and other parameters are as defined in Theorem 2.*

*Proof.* Without loss of generality, we use $c$ serve as a generic constant since we focus on the order in terms of the sample number and thus omit the constant factor.

$$\begin{aligned}
\varepsilon_P(\hat{h}_Q^*) &= \mathcal{R}_P(\hat{h}_Q^*) - \mathcal{R}_P(h_P^*) \\
&\leq \hat{\mathcal{R}}_P(\hat{h}_Q^*) - \hat{\mathcal{R}}_P(h_P^*) + c\sqrt{\min\{P(\hat{h}_Q^* \neq h_P^*), \hat{P}(\hat{h}_Q^* \neq h_P^*)\} A(n_T, \delta)} + c A(n_T, \delta) \\
&\leq c\sqrt{\varepsilon_P^{\beta_P}(\hat{h}_Q^*) A(n_T, \delta)} + c A(n_T, \delta).
\end{aligned}$$

The first inequality is due to Lemma 1 and second inequality follows from Lemma 2 and Noise assumption 1. Then we have the following result, which completes the proof:

$$\varepsilon_P(\hat{h}_Q^*) \leq c A(n_T, \delta)^{\frac{1}{2 - \beta_P}}.$$

$\square$

**Lemma 2.** *If $\hat{\mathcal{R}}_P(\hat{h}_Q^*) \leq \hat{\mathcal{R}}_P(h_Q^*)$, with probability at least $1 - \delta$,*

$$\hat{\mathcal{R}}_P(\hat{h}_Q^*) - \hat{\mathcal{R}}_P(h_P^*) = \varepsilon_P(h_Q^*) + \mathcal{O}(A(n_T, \delta)).$$

*Proof.*

$$\hat{\mathcal{R}}_P(\hat{h}_Q^*) - \hat{\mathcal{R}}_P(h_P^*) \leq \hat{\mathcal{R}}_P(h_Q^*) - \hat{\mathcal{R}}_P(h_P^*)$$

$$\leq \mathcal{R}_P(h_Q^*) - \mathcal{R}_P(h_P^*) + c\sqrt{\min\{P(h_Q^* \neq h_P^*), \hat{P}(h_Q^* \neq h_P^*)\}A(n_T,\delta)} + cA(n_T,\delta)$$

$$= \varepsilon_P(h_Q^*) + \mathcal{O}(A(n_T,\delta)).$$

$\square$

**Theorem 4** (Convergence Rate for SAFARI ). *Under Assumptions 2 and 3, let constant learning rate $\eta$ satisfy $(\frac{1}{2} - 4LK\eta - 20K(L + 4KL^3\eta)\eta^2) > 0$. Then, the sequence $\{\mathbf{x}_t\}$ generated by the SAFARI algorithm satisfies:*

$$\frac{1}{T}\sum_{t=0}^{T-1} \mathbb{E}\|\nabla F(\mathbf{x}_t)\|^2 \leq \frac{1}{c}\left[\frac{F(\mathbf{x}_0) - F(\mathbf{x}^*)}{\eta K T}\right] + \frac{1}{c}\left[\left(5KL\eta^2 + 20K^2L^3\eta^3 + 2L\eta\right)\sigma^2\right]$$

$$+ \frac{1}{c}\left[\left(\frac{1}{K^2} + \frac{L\eta}{K}\right)\frac{1}{T}\sum_{t=0}^{T-1}c_t^2\right],$$

*where $c$ is a constant and $\mathbf{x}^*$ denotes an optimal solution.*

*Proof.* Let $\bar{\Delta}_t = \frac{1}{|S_t|}\sum_{i \in S_t}\hat{\Delta}_t^i, \bar{\mathbf{g}}_t = \Delta_t^0 + \frac{1}{|S_t|}\sum_{i \in S_t}\hat{\Delta}_t^i = \Delta_t^0 + \bar{\Delta}_t.$

$$\mathbb{E}_t[F(\mathbf{x}_{t+1})] \leq F(\mathbf{x}_t) + \langle\nabla F(\mathbf{x}_t), \mathbb{E}_t[\mathbf{x}_{t+1} - \mathbf{x}_t]\rangle + \frac{L}{2}\mathbb{E}_t[\|\mathbf{x}_{t+1} - \mathbf{x}_t\|^2]$$

$$= F(\mathbf{x}_t) + \langle\nabla F(\mathbf{x}_t), \eta\mathbb{E}_t\bar{\mathbf{g}}_t\rangle + \frac{L}{2}\eta^2\mathbb{E}_t[\|\bar{\mathbf{g}}_t\|^2]$$

$$= F(\mathbf{x}_t) - \eta K\|\nabla F(\mathbf{x}_t)\|^2 + \underbrace{\langle\nabla F(\mathbf{x}_t), \eta K\nabla F(\mathbf{x}_t) + \eta\mathbb{E}_t[\Delta_t^0 + \bar{\Delta}_t]\rangle}_{A_1} + \underbrace{\frac{L}{2}\eta^2\mathbb{E}_t[\|\Delta_t^0 + \bar{\Delta}_t\|^2]}_{A_2}.$$

$$A_1 = \langle\nabla F(\mathbf{x}_t), \eta K\nabla F(\mathbf{x}_t) + \eta\mathbb{E}_t[\Delta_t^0 + \bar{\Delta}_t]\rangle = \eta K\langle\nabla F(\mathbf{x}_t), \nabla F(\mathbf{x}_t) + \frac{1}{K}\mathbb{E}_t[\Delta_t^0 + \bar{\Delta}_t]\rangle$$

$$\leq \frac{1}{2}\eta K\|\nabla F(\mathbf{x}_t)\|^2 + \frac{1}{2}\eta K\|\nabla F(\mathbf{x}_t) + \frac{1}{K}\mathbb{E}_t[\Delta_t^0 + \bar{\Delta}_t]\|^2$$

Note that $\Delta_t^0 = -\sum_{k=0}^{K-1}\nabla F(\mathbf{x}_{t,k}^0, \xi_{t,k}^0)$. We have

$$\frac{1}{2}\eta K\|\nabla F(\mathbf{x}_t) + \frac{1}{K}\mathbb{E}_t[\Delta_t^0 + \bar{\Delta}_t]\|^2 \leq \eta K\|\nabla F(\mathbf{x}_t) + \frac{1}{K}\mathbb{E}_t[\Delta_t^0]\|^2 + \eta K\|\frac{1}{K}\mathbb{E}_t[\bar{\Delta}_t]\|^2$$

$$\leq \eta K\|\nabla F(\mathbf{x}_t) - \mathbb{E}_t\left[\frac{1}{K}\sum_{k=0}^{K-1}\nabla F(\mathbf{x}_{t,k}^0, \xi_{t,k}^0)\right]\|^2 + \frac{\eta}{K}\mathbb{E}_t\|[\bar{\Delta}_t]\|^2$$

$$= \eta K\|\nabla F(\mathbf{x}_t) - \frac{1}{K}\sum_{k=0}^{K-1}\nabla F(\mathbf{x}_{t,k}^0)\|^2 + \frac{\eta}{K}\mathbb{E}_t\|\frac{1}{|S_t|}\sum_{i \in S_t}\hat{\Delta}_t^i\|^2$$

$$\leq \eta\sum_{k=0}^{K-1}\|\nabla F(\mathbf{x}_t) - \nabla F(\mathbf{x}_{t,k}^0)\|^2 + \eta\frac{1}{|S_t|}\mathbb{E}_t\sum_{i \in S_t}\|\hat{\Delta}_t^i\|^2$$

$$\leq \eta L\sum_{k=0}^{K-1}\|\mathbf{x}_t - \mathbf{x}_{t,k}^0\|^2 + \frac{\eta}{K}c_t^2.$$

So, we can bound $A_1$ as following:

$$A_1 \leq \frac{1}{2}\eta K\|\nabla F(\mathbf{x}_t)\|^2 + \eta L\sum_{k=0}^{K-1}\|\mathbf{x}_t - \mathbf{x}_{t,k}^0\|^2 + \frac{\eta}{K}c_t^2.$$

$$A_2 = \frac{L}{2}\eta^2 \mathbb{E}_t \left[\|\Delta_t^0 + \bar{\Delta}_t\|^2\right] \leq L\eta^2 \mathbb{E}_t \left[\|\Delta_t^0\|^2\right] + L\eta^2 \mathbb{E}_t \left[\|\bar{\Delta}_t\|^2\right].$$

$$\mathbb{E}_t \left[\|\Delta_t^0\|\right] = \|\sum_{k=0}^{K-1} \nabla F(\mathbf{x}_{t,k}^0, \xi_{t,k}^0)\|^2$$

$$\leq 2\|\sum_{k=0}^{K-1} \nabla F(\mathbf{x}_{t,k}^0)\|^2 + 2K\sigma^2$$

$$\leq 2\|\sum_{k=0}^{K-1} \left[\nabla F(\mathbf{x}_{t,k}^0) - \nabla F(\mathbf{x}_t) + \nabla F(\mathbf{x}_t)\right]\|^2 + 2K\sigma^2$$

$$\leq 4K\sum_{k=0}^{K-1} \left[\|\nabla F(\mathbf{x}_{t,k}^0) - \nabla F(\mathbf{x}_t)\|^2 + \|\nabla F(\mathbf{x}_t)\|^2\right] + 2K\sigma^2$$

$$\leq 2K\sigma^2 + 4KL^2 \sum_{k=0}^{K-1} \|\mathbf{x}_{t,k}^0 - \mathbf{x}_t\|^2 + 4K^2\|\nabla F(\mathbf{x}_t)\|^2,$$

where the first inequality is due to assumption 1 and $\{\nabla F(\mathbf{x}_{t,k}^0, \xi_{t,k}^0) - \nabla F(\mathbf{x}_{t,k}^0)\}$ form a martingale difference sequence (see Lemma 4 in [4]).

Hence, we can bound $A_2$ as following:

$$A_2 \leq 2KL\eta^2\sigma^2 + 4KL^3\eta^2 \sum_{k=0}^{K-1} \|\mathbf{x}_{t,k}^0 - \mathbf{x}_t\|^2 + 4LK^2\eta^2\|\nabla F(\mathbf{x}_t)\|^2 + L\eta^2 c_t^2.$$

Plugging the bound of $A_1$ and $A_2$ into the smoothness inequality, we have:

$$\mathbb{E}_t[F(\mathbf{x}_{t+1})] \leq F(\mathbf{x}_t) - \eta K\|\nabla F(\mathbf{x}_t)\|^2 + \underbrace{\left\langle \nabla F(\mathbf{x}_t), \eta K \nabla F(\mathbf{x}_t) + \eta \mathbb{E}_t \left[\Delta_t^0 + \bar{\Delta}_t\right]\right\rangle}_{A_1}$$

$$+ \underbrace{\frac{L}{2}\eta^2 \mathbb{E}_t \left[\|\Delta_t^0 + \bar{\Delta}_t\|^2\right]}_{A_2}$$

$$\leq F(\mathbf{x}_t) - \eta K(\frac{1}{2} - 4LK\eta)\|\nabla F(\mathbf{x}_t)\|^2 + \left(\eta L + 4KL^3\eta^2\right) \sum_{k=0}^{K-1} \mathbb{E}_t[\|\mathbf{x}_{t,k}^0 - \mathbf{x}_t\|^2]$$

$$+ \left(\frac{\eta}{K} + L\eta^2\right) c_t^2 + 2KL\eta^2\sigma^2.$$

For the server, we have the following results for the norm of parameter changes for one local computation:

$$\mathbb{E}[\|\mathbf{x}_{t,k}^0 - \mathbf{x}_t\|^2] = \mathbb{E}[\|\mathbf{x}_{t,k-1}^0 - \mathbf{x}_t - \eta g_{t,k-1}^0\|^2]$$

$$= \mathbb{E}\left[\|\mathbf{x}_{t,k-1}^0 - \mathbf{x}_t - \eta \nabla F(\mathbf{x}_{t,k-1}^0)\|^2\right] + \mathbb{E}\|\eta\left(g_{t,k-1}^0 - \nabla F(\mathbf{x}_{t,k-1}^0)\right)\|^2$$

$$= (1 + \frac{1}{2K-1})\mathbb{E}\left[\|\mathbf{x}_{t,k-1}^0 - \mathbf{x}_t\|^2\right] + \mathbb{E}\|\eta\left(g_{t,k-1}^0 - \nabla F(\mathbf{x}_{t,k-1}^0)\right)\|^2$$

$$+ 2K\mathbb{E}\|\eta \nabla F(\mathbf{x}_{t,k-1}^0) - \eta \nabla F(\mathbf{x}_t) + \eta \nabla F(\mathbf{x}_t)\|^2$$

$$= (1 + \frac{1}{2K-1})\mathbb{E}\left[\|\mathbf{x}_{t,k-1}^0 - \mathbf{x}_t\|^2\right] + \mathbb{E}\|\eta\left(g_{t,k-1}^0 - \nabla F(\mathbf{x}_{t,k-1}^0)\right)\|^2$$

$$+ 4K\eta^2\|\nabla F(\mathbf{x}_{t,k-1}^0) - \nabla F(\mathbf{x}_t)\|^2 + 4K\eta^2\|\nabla F(\mathbf{x}_t)\|^2$$

$$\leq (1 + \frac{1}{2K-1} + 4KL^2\eta^2)\mathbb{E}\left[\|\mathbf{x}_{t,k-1}^0 - \mathbf{x}_t\|^2\right] + \eta^2\sigma^2 + 4K\eta^2\|\nabla F(\mathbf{x}_t)\|^2$$

$$\leq (1 + \frac{1}{K-1})\mathbb{E}\left[\|\mathbf{x}_{t,k-1}^0 - \mathbf{x}_t\|^2\right] + \eta^2\sigma^2 + 4K\eta^2\|\nabla F(\mathbf{x}_t)\|^2.$$

Unrolling the recursion, we obtain the following:

$$\mathbb{E}[\|\mathbf{x}_{t,k}^0 - \mathbf{x}_t\|^2] = \sum_{p=0}^{k-1}(1+\frac{1}{K-1})^p\left(\eta^2\sigma^2 + 4K\eta^2\|\nabla F(\mathbf{x}_t)\|^2\right)$$

$$\leq (K-1)\left[\left(1+\frac{1}{K-1}\right)^K - 1\right]\left(\eta^2\sigma^2 + 4K\eta^2\|\nabla F(\mathbf{x}_t)\|^2\right)$$

$$\leq 5K\eta^2\sigma^2 + 20K^2\eta^2\|\nabla F(\mathbf{x}_t)\|^2.$$

Putting the pieces together, we obtain

$$\mathbb{E}_t[F(\mathbf{x}_{t+1})] - F(\mathbf{x}_t) \leq -\eta K(\frac{1}{2} - 4LK\eta - 20K(L + 4KL^3\eta)\eta^2)\|\nabla F(\mathbf{x}_t)\|^2$$

$$+ \eta K\left(5KL\eta^2 + 20K^2L^3\eta^3 + 2L\eta\right)\sigma^2 + \left(\frac{\eta}{K} + L\eta^2\right)c_t^2$$

$$\leq -c\eta K\|\nabla F(\mathbf{x}_t)\|^2 + \eta K\left(5KL\eta^2 + 20K^2L^3\eta^3 + 2L\eta\right)\sigma^2$$

$$+ \left(\frac{\eta}{K} + L\eta^2\right)c_t^2.$$

The last inequality follows from that there exist such constant $c$ if $(\frac{1}{2} - 4LK\eta - 20K(L + 4KL^3\eta)\eta^2) > 0$.

Summing over $t = 0$ to $T-1$, we have

$$\frac{1}{T}\sum_{t=0}^{T-1}\mathbb{E}\|\nabla F(\mathbf{x}_t)\|^2 \leq \frac{1}{c}\left[\frac{F(\mathbf{x}_0) - F(\mathbf{x}^*)}{\eta K T} + \left(5KL\eta^2 + 20K^2L^3\eta^3 + 2L\eta\right)\sigma^2\right.$$

$$\left. + \left(\frac{1}{K^2} + \frac{L\eta}{K}\right)\frac{1}{T}\sum_{t=0}^{T-1}c_t^2\right]$$

$$\square$$

# B   Experiments

In this section, we provide the details of the numerical experiments and some additional experimental results.

## B.1   Models and Datasets

We test the SAFARI algorithm by running two models on two different types of datasets, including 1) multinomial logistic regression (LR) on MNIST, and 2) convolutional neural network (CNN) on CIFAR-10. Both datasets are chose from a previous FL paper [1], and they are now widely used as benchmarks for FL research [7, 33].

MNIST and CIFAR-10 have ten classes of images separately. In order to impose the heterogeneity of the data, we partition the dataset according to the number of classes ($p$) that each client contains. We distribute these data to $M = 10$ clients, and each client only has a certain number of classes. Specifically, each client randomly selects $p$ classes of images and then evenly samples training and test data-points within these $p$ classes of images without replacement. For example, if $p = 2$, each client only samples training and test data-points within two classes of images, which causes the heterogeneity among different clients. If $p = 10$, each client contains training and test samples that selects from ten classes. This situation is almost the same as i.i.d. case. Hence, the number of classes ($p$) in each client's local dataset can be used to represent the level of non-i.i.d. qualitatively. In addition, to mimic incomplete client participation, we enforce $s$ clients to be exempt from participation, where the index $s$ can be used to represent the degree of incomplete client participation. Specifically, we

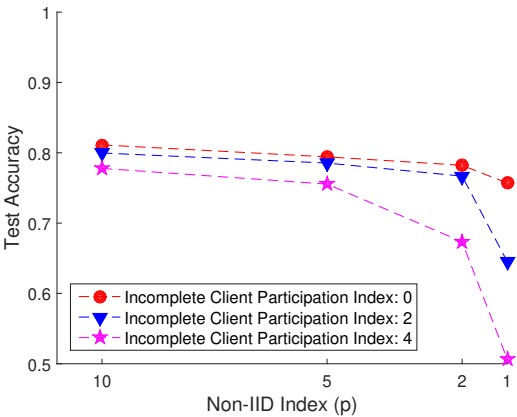

Figure 2: Test Accuracy of FedAvg on CIFAR-10 with incomplete client participation. Larger incomplete client participation index means less clients participate in the training, and smaller non-i.i.d. index means the data across clients is more heterogeneous.

assume there are $M = 10$ clients in total, and $m = 5$ clients participate in each communication round. These clients are uniformly sampled from $M - s$ clients. Larger incomplete client participation index $s$ means less clients participate in the training.

For both MNIST and CIFAR-10, the learning rate is 0.1, and the local epoch is 1. For MNIST, the batch size is 64, and the total communication round is 150. For CIFAR-10, the batch size is 500, and the total communication round is 4000. To simulate the data heterogeneity, we use $p = [10, 5, 2, 1]$ as a proxy to represent the degree of non-i.i.d. on MNIST and CIFAR-10 datasets. To emulate the effect of incomplete client participation, we set $s = [0, 2, 4]$ to represent the degree of incomplete client participation for the SAFARI algorithm, the FedAvg algorithm, and the SGD algorithm. Last two algorithms are employed as the baselines to compare with our algorithm. The hyper-parameter $c_t$ in the SAFARI algorithm is set to 0.1 both on MNIST and CIFAR-10. To compare the effect of the collaboration from server, we add $[50, 100, 500, 1000]$ data to the server's side for MNIST and $[500, 1000, 5000]$ for CIFAR-10.

## B.2 Additional Experimental Results

In Figure B.2, we show the test accuracy of FedAvg algorithm on CIFAR-10 for different Non-IID index $p$ and incomplete client participation index $s$. In the case of $p = 10$, the test accuracy of $s = 4$ and $s = 0$ is not much different whereas the test accuracy of $s = 4$ is 25% lower than that of $s = 1$ in the case of $p = 1$. Incomplete client participation has no impact on the performance for nearly homogeneous data, but it causes catastrophical performance degradation for highly Non-IID data.

In Figure 3, we show the test accuracy of the SAFARI algorithm, the FedAvg algorithm, and the SGD algorithm on MNIST for incomplete client participation $s = 4$ and different Non-IID index $p$. The evidences of the observations are provided visually as follows:

- Compared to FedAvg in the case of $p = 1$ (see Figure 3(d)), with only 50 data at server's side (0.1% of the total training data), there is a non-negligible increase of test accuracy for our SAFARI algorithm. This increase increases as more data is added to the server's side.

- In nearly homogeneous case when $p = 5$ or $p = 10$ (see Figure 3(a) and 3(b)), there is actually no improvement of the test accuracy with these auxiliary data added to the server's side, comparing SAFARI with FedAvg.

- Compared SAFARI with SGD (for centralized learning solely on server's data) in nearly homogeneous case when $p = 5$ or $p = 10$ (see Figure 3(a) and 3(b)), the collaborations from clients significantly improves the performance, especially with less data on the server's side.

- In highly heterogeneous case when $p = 2$ or $p = 1$ (see Figure 3(c) and 3(d)), it shows no obvious improvement from the collaboration of clients comparing SAFARI to SGD.

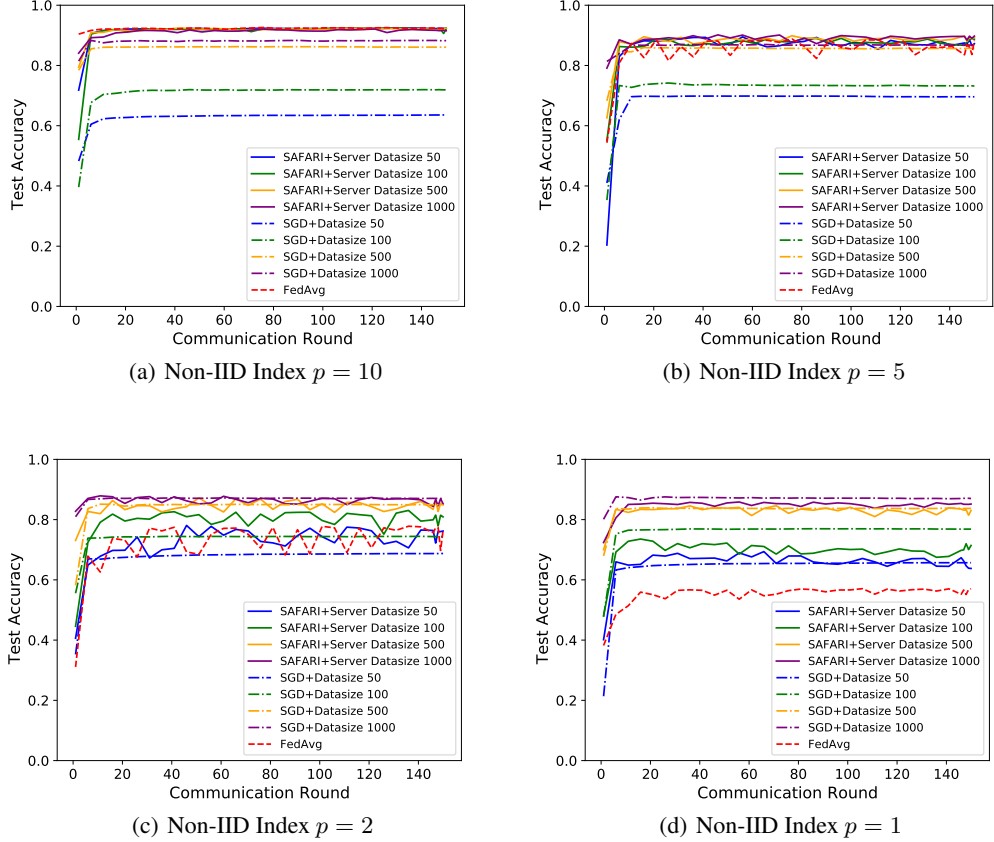

Figure 3: Test accuracy of SAFARI , FedAvg, and SGD algorithm on MNIST with incomplete client participation $s = 4$ and different Non-IID index $p$. Smaller $p$ means the data across clients is more heterogeneous.

Table 1: Test accuracy improvement (%) for SAFARI compared with FedAvg on CIFAR-10 with incomplete client participation $s = 4$. '-' means no statistical difference within $2\%$ error bar.

| Server Datasize | Non-IID Index ($p$) | | | |
| --- | --- | --- | --- | --- |
| | 10 | 5 | 2 | 1 |
| 500 | - | - | - | - |
| 1000 | - | - | - | 3.55 |
| 5000 | - | - | 5.45 | 16.08 |

In Table 1, we show the comparison between our SAFARI algorithm and FedAvg algorithm on CIFAR-10 for incomplete client participation $s = 4$. The observations are further illustrated: 1) There is non-negligible increase of the test accuracy for SAFARI algorithm with small amount of auxiliary data at server's side. With 5000 data at server's side, the test accuracy increases by 16.08%. 2) There is actually no improvement with these auxiliary data for nearly homogeneous case (e.g., $p = 10$ or $p = 5$), which is denoted by '-' in the table.

In Table 2, we show the difference between our SAFARI algorithm and SGD, which is for centralized learning solely on server's data, for incomplete client participation $s = 4$ on CIFAR-10. When the size of data on server's side is small, the collaborations from clients significantly improve the performance of the SAFARI algorithm. Even in the highly heterogeneous case when $p = 1$, the test accuracy can be improved by 10% for only 500 data on the server's side (0.8% of the total training data). This observation further validates our theoretical analysis in Theorem 2.

Table 2: Test accuracy improvement (%) of SAFARI under incomplete client participation $s = 4$ compared with SGD in centralized learning on CIFAR-10. Smaller Non-IID index means the data across clients is more heterogeneous.

| SERVER DATASIZE | NON-IID INDEX ($p$) | | | |
|---|---|---|---|---|
| | 10 | 5 | 2 | 1 |
| 500 | 35.67 | 33.48 | 27.60 | 10.77 |
| 1000 | 31.23 | 28.46 | 22.36 | 7.62 |
| 5000 | 13.99 | 11.11 | 7.88 | 3.40 |

