# OpenReview forum: "With a Little Help from My Friend: Server-Aided Federated Learning with Partial Client Participation"
_NeurIPS.cc/2022/Workshop/Federated_Learning — FL-NeurIPS 2022 Poster_

### Official Review · Reviewer_nvzt · 2022-10-16
**Simple idea with limited novelty and strong assumptions**

The paper studies the problem of partial/incomplete client participation in FL. The authors first show that FL with incomplete participation is  not PAC-learnable. Then they propose an algorithm SAFARI, that deals with this with this incomplete participation using an auxiliary dataset at the server.

**Pros**

* The idea of studying the effect of partial participation in FL through the lens of PAC learnability is interesting.

* The authors point out some interesting connections between their proposed algorithm and the domain adaption problem.


**Cons**

* The result in Theorem 1 seems a bit trivial. Essentially, if some clients are never going to participate then some domains of the joint distribution $P$ are never going to be sampled. Thus, it is easy to see why the generalization error of any learner will non-zero with constant probability. I would also suggest the authors to omit the proof sketch included below Theorem 1 since it currently does not add anything significant to the discussion.

* The assumption about having an auxiliary dataset at the server is quite strong. Firstly even if the server has access to such auxiliary data, assuming that the data is iid sampled from the joint distribution $P$ is unrealistic. Also, in the works that the authors cite, this auxiliary dataset is mostly used for validation purposes rather than training.

* Again definition/assumption 4 seems quite strong. In my understanding, this implies
$ \epsilon_{P}[h_{Q}^*] = 0$ since $\epsilon_{Q}[h_Q^*] = 0$. If so, the authors should explicitly mention this as a consequence of their definition.

* $F$ is not defined in Section 3 which makes the algorithm/convergence confusing. Regardless of the definition of $F$, it seems that the server can compute an unbiased estimate of the gradient of $F$. This again makes the algorithm convergence trivial and unrelated to FL. For instance it seems you can set all the $c_t = 0$ and still get convergence, meaning you are not utilizing any work done by the clients.

**General Comments:**

I appreciate the authors attempt at trying to deal with incomplete client participation in FL. However, I feel in the case that some clients never participate, the situation is quite hopeless and requires strong assumptions such as the auxiliary dataset to show any guarantees. Maybe a relaxed version would be to consider all clients participate at least once or participate very infrequently as done in the following works.

[1] Wang, Shiqiang, and Mingyue Ji. "A Unified Analysis of Federated Learning with Arbitrary Client Participation." arXiv preprint arXiv:2205.13648 (2022).

[2] Yang, Haibo, et al. "Anarchic federated learning." International Conference on Machine Learning. PMLR, 2022.

I would also suggest rethinking the paper title since it currently a bit wordy and does not carry the right impact.

**Typos**

Line 24: "resulted from" -> "resulting from"

Line 128: "distribution derivations" -> "distribution deviations"?

---

### Official Review · Reviewer_upF1 · 2022-10-17

**Summary**:
This paper investigates the problem of incomplete client participation by providing theoretical analysis of the generalization error when not all clients participate to train the global model of FL, showing that the error may never go to 0 due to this incomplete participation. With this observation, the paper proposes the SA-FL framework which is the server having a separate dataset that is assumed to be sampled from the union of the clients' local data's underlying distribution $P_i,i\in[M]$ which enables the generalization error to go to 0 under certain assumptions. The work also proposes an algorithm SAFARI that uses the SA-FL framework and provide convergence guarantees for this algorithm.

**Strength**:
- Analyzing the problem of incomplete client participation through the lens of PAC learnability is interesting.
- Provides convergence guarantees for the proposed SAFARI algorithm.

**Weaknesses**:
- The framework SA-FL is analyzed under a rather restrictive setting in that the aggregation weights (either $\alpha_i$ or $\lambda_i$ are all $1/M$).
- Although the authors argue that the Server having the auxiliary dataset is a reasonable assumption, I do not agree with this because the
auxiliary dataset that the SA-FL framework points to is a dataset that is consisted of datasamples sampled from the union of the clients' local data's underlying distribution $P_i,i\in[M]$. In order for this assumption to be satisfied, the server needs to know all of the clients' local data's underlying distributions, and this contradicts the underlying motivation for doing FL in the first place; ensuring clients' data privacy.
- While the paper claims to look at incomplete client participation, the actual SAFARI algorithm doesn't take this in to account. The steps are almost identical to those of the standard FL algorithm; $m$ clients are sampled and they have to send their update back. Also, what and how we define $c_t$ in the SAFARI algorithm is not clear.

**Decision**:
- Overall, although the work looks into an interesting idea, I believe the work only weakly address this idea with the strong assumptions and weak algorithm. I suggest the authors relax these assumptions and give more validation to the algorithm or improve it to properly address the problem of incomplete client participation.

---

### Official Review · Reviewer_WYvL · 2022-10-18

This paper tackles the problem of Federated Learning (FL) over clients with heterogeneous data and incomplete client participation. The authors show that in this setting FL is not PAC-learnable, and propose to address this by considering an IID dataset at the server and some assumptions on how the distribution of sampled clients + the server dataset relates to the target distribution. They propose a simple variation of FedAvg where the server also performs local updates on its dataset.

The paper certainly addressed an interesting problem. The fact that FL is not PAC-learnable under incomplete participation is not surprising (we cannot leatn what we don't see), Theorem 1 is moderately interesting but has the merit of formally establishing this claim.

I see two main limitations of the approach and results:
- Assuming an IID auxiliary dataset at the server (even small) is not very realistic. One of the advantages of FL is precisely to learn models on actual user data, whose global distribution is very difficult to estimate otherwise.
- I am not sure that Theorem 2 says much about the usefulness of the auxiliary dataset. Indeed, the authors assume $n_T << n_S$, hence even if one takes $n_T=0$ (no auxiliary data) then the generalization bound is essentially the same. This result is in fact mostly due to assuming that Definition 4 holds with large enough $\beta$ (which is a quite strong assumption). In other words, Theorem 2 gives a condition relating $\mathcal D$ (the empirical client distribution) to $P$ (the true client distribution) under which learning on $\mathcal D$ leads to good performance on $P$. The auxiliary data aspect thus appears incidental.

Despite the above limitations, I think the paper can yield interesting discussions at the workshop so I recommend to accept it if there is space.

Typo: line 148, Definition 1 should be Definition 4.

---

### Decision · Program_Chairs · 2022-10-20

Accept (Poster)